# Anaesthetists' attitudes towards attending the funerals of their patients: A cross-sectional study among Australian and New Zealand anaesthetists

Kwangtaek Kim[1], Leonid Churilov[2], Chong Oon Tan[1], Tuong Phan[3], Jake Geertsema[4], Roni Krieser[5], Rishi Mehra[6], Paul Anthony Stewart[7], Clive Rachbuch[8], Andrew Huang[1], Laurence Weinberg[1,9]*

1 Department of Anaesthesia, Austin Hospital, Heidelberg, Victoria, Australia, 2 Department of Medicine (Austin Health) & Melbourne Brain Centre at Royal Melbourne Hospital, Melbourne Medical School, Victoria, Australia, 3 Department of Anaesthesia, St Vincent's Hospital, Fitzroy, Victoria, Australia, 4 Department of Anaesthesia, The Northern Hospital, Epping, Victoria, Australia, 5 Department of Anaesthesia, The Royal Melbourne Hospital, Parkville, Victoria, Australia, 6 Department of Anaesthesia, The Alfred Hospital, Melbourne, Victoria, Australia, 7 Department of Anaesthesia, Sydney Adventist Hospital, Wahroonga, New South Wales, Australia, 8 Department of Anaesthesia, Box Hill Hospital, Box Hill, Victoria, Australia, 9 Department of Surgery, Austin Health, The University of Melbourne, Victoria, Australia

* laurence.weinberg@austin.org.au

**Data Availability Statement:** There are no restrictions on our data access and we have now

## Abstract

A patient's death can pose significant stress on the family and the treating anaesthetist. Anaesthetists' attitudes about the benefits of and barriers to attending a patient's funeral are unknown. Therefore, we performed a prospective, cross-sectional study to ascertain the frequency of anaesthetists' attendance at a patient's funeral and their perceptions about the benefits and barriers. The primary aim was to investigate the attitudes of anaesthetists towards attending the funeral of a patient. The secondary aims were to examine the perceived benefits of and barriers to attending the funeral and to explore the rate of bonds being formed between anaesthetists, patients and families. Of the 424 anaesthetists who completed the survey (response rate 21.2%), 25 (5.9%) had attended a patient's funeral. Of the participants, 364 (85.9%) rarely formed special bonds with patients or their families; 233 (55%) believed that forming a special bond would increase the likelihood of their attendance. Showing respect to patients or their families was the most commonly perceived benefit of attending a funeral. Participants found expression of personal grief and caring for the patient at the end-of-life and beyond beneficial to themselves and the family. Fear of their attendance being misinterpreted or perceived as not warranted by the family as well as time restraints were barriers for their attendance. Most anaesthetists had never attended a patient's funeral. Few anaesthetists form close relationships with patients or their families. Respect, expression of grief and caring beyond life were perceived benefits of attendance. Families misinterpreting the purpose of attendance or not expecting their attendance and time restraints were commonly perceived barriers.

**Trial registration:** ACTRN 12618000503224.

uploaded our anonymized data set as a Supporting Information file.

**Funding:** The author(s) received no specific funding for this work.

**Competing interests:** The authors have declared that no competing interests exist.

## Introduction

A patient's death can impose significant stress on both the family and the treating team [1, 2]. When faced with such a stressful event, clinicians frequently develop strategies to minimise the impact of the event to both themselves and the family. Examples include being available to answer the family's questions [3] and making a phone call or writing a condolence letter to the family [4]. Although not a frequent practice, medical practitioners may also attend the funeral of their patients. Current literature mainly presents anecdotal experiences and individual opinions of clinicians regarding attendance at a patient's funeral [1, 5–19]. Objective data about medical practitioners' attitudes towards attending a patient's funeral and the frequency, reason and outcome of attendance are only available for a limited number of specialties, namely palliative care [3, 20, 21], oncology [3, 20–23] and paediatrics [2, 24–27].

With an increasing focus on perioperative medicine in anaesthesia training, anaesthetists are increasingly responsive to the preferences, needs and values of the individual patient. With more anaesthetists embracing perioperative medicine as a distinct subspecialty, a deeper understanding of what is important to the patient is being developed. This understanding fosters trust, establishes mutual respect and facilitates a unique clinician–patient-centred approach to shared planning and decision-making. Hence, anaesthetists currently have more opportunities to develop a strong rapport with patients and their families. Accordingly, over time, more anaesthetists will either be interested or be asked to attend the funeral of their patients. However, the attitudes of other medical specialists towards funeral attendance remains largely unexplored, and there is no large study specifically exploring the attitudes, benefits and barriers of attending a patient's funeral as perceived by anaesthetists [28]. Two existing studies have partly explored anaesthetists' attitudes towards attending a patient's funeral. However, these had significant limitations in that they were not representative of a large anaesthetist population. In an Australian study, only five participants were anaesthetists [20]. Similarly, in an American study, 22 paediatric critical care specialists were subspecialised in anaesthesia [27]. Insufficient information about the prevalence, rationale and attitudes of anaesthetists' attendance at a patient's funeral means that these findings are not generalisable to the broader Australian anaesthesia community.

Therefore, we conducted a cross-sectional, mixed-methods study to fill this knowledge gap. Our primary aims were to ascertain the attitudes of Australian and New Zealand anaesthetists towards attending a patient's funeral and the perceived benefits of and barriers to attending these funerals. The secondary aims were to examine the perceived benefits of and barriers to attending the funeral and to explore the rate of bonds being formed between anaesthetists, patients and families. We also examined if a patient's unexpected death influences anaesthetists' attendance at the funeral.

## Methods

### Study population

This study was approved by the Human Research Ethics Committee (LNR/17/Austin/422) and registered with the Australian and New Zealand College of Anaesthetists (ANZCA) (Trial number: ACTRN 12618000503224; website address: https://www.anzctr.org.au/Trial/Registration/TrialReview.aspx?id=374549&showOriginal=true&isReview=true). We conducted a prospective, mixed-methods survey of practising Fellow anaesthetists in Australia and New Zealand registered with the ANZCA. Anaesthesia registrars and trainees and retired Fellows were excluded.

## Study design

After a literature review of anaesthetists' views on attending a patient's funeral, we designed and developed an online survey using commercial software (SurveyMonkey Inc., San Mateo, California, USA). The survey was pilot tested on 10 anaesthetists from rural, secondary and tertiary level hospitals to verify whether the survey questions were comprehensible, appropriate, well-defined and not misleading. We asked whether the questions were presented in a consistent manner. Responses from the pilot testing did not lead to reformulation of any questions. No additional questions were included and no questions were excluded. Minor corrections to syntax and grammar were made, and the final survey questions differed only slightly from the pilot questionnaire. The final survey consisted of 17 questions. Given that open and reflective discussions about the study, its aims, objectives and design occurred between the researchers and the pilot study participants, a tendency towards a more prejudiced viewpoint could have been introduced; therefore, to avoid this bias, the pilot study participants were excluded from participating in the final survey.

The first 10 questions explored participants' previous experience of attending a patient's funeral, frequency of formation of special bonds with patients or their families, factors that affect their attendance at a patient's funeral and perceived benefits of and barriers to attending a patient's funeral. Having a special bond with patients or their families and the unexpected death of a patient were the two factors that were investigated in the survey. A 5-point Likert scale (strongly agree to strongly disagree) was used to determine if these two factors affected participants' attendance at their patient's funeral. Survey questions that explored perceived benefits of and barriers to attending a patient's funeral allowed participants to choose none, one or multiple options. There were also three open-ended questions where participants could provide free-text responses regarding the benefits of and barriers to their attendance at a patient's funeral. The next seven questions explored participants' geographic region, age, gender, type of hospital (rural or urban), their area of practice as well as whether they mainly practise in a public or private setting. A copy of the survey is presented in S1 Appendix.

## Survey distribution

An invitation email was sent directly to the directors of anaesthesia departments in Australian and New Zealand hospitals listed on the ANZCA website (http://www.anzca.edu.au). This invitation email included the objectives and a summary of the study, written information about consent (which stated that consent was assumed on completion of the survey as participation was completely voluntary) and a link to the online survey. A copy of the invitation email is presented in S2 Appendix. The participant information and consent form are presented in S3 Appendix.

Based on information from the ANZCA, we sent the survey to Australian hospitals in the following states and territories: New South Wales (NSW, n = 34), Victoria (VIC, n = 19), Queensland (QLD, n = 19), Western Australia (WA, n = 8), South Australia (SA, n = 4), Tasmania (TAS, n = 3), the Northern Territory (NT, n = 2) and the Australian Capital Territory (ACT, n = 2). We also sent the survey to 17 New Zealand (NZ) hospitals. Where possible, we estimated the number of full- and part-time anaesthetists practising in each location based on direct information from the directors. Then, we corrected for the number of part-time anaesthetists who frequently worked in more than one public hospital. We estimated that up to 2000 anaesthesia Fellows received the survey.

Directors were asked to forward the invitation email to consultant anaesthetists in their department. Distribution of the email by the directors and participation in the survey were completely voluntary. A total of 110 hospitals were contacted between February and March

2018. Two directors chose not to participate in the survey. The survey was open from February–April 2018. During this time, participants were able to access and complete the survey at their convenience. Participants could only complete the survey once, negating any risk of response duplication. No reminder emails were sent. No compensation was offered for participation in the study. All responses were completely anonymous, and no Internet Protocol (IP) addresses were collected.

### Data analysis

Where possible, we obtained data from the ANZCA for the demographic questions pertaining to currently registered anaesthetists. Data for age distribution, gender, country of practice and location of current practice for comparative purposes were obtained. Statistical analysis was performed using commercial statistical software STATA/IC v.13 with a p value of 0.05 to indicate statistical significance. The association between participants' characteristics and their responses to specific questions was investigated using Fisher's exact test, binary logistic regression for dichotomous outcomes and Poisson regression for count outcomes. Corresponding effect sizes were reported as either Odds Ratios (ORs) or Incidence Rate Ratios (IRRs) as appropriate with respective 95% confidence intervals. To extract the main themes from free-text responses, NVivo v.12 was employed. This powerful and automated process supported a structured analysis by automatically coding the sets of data. In turn, we gained deeper insights from the data by being able to automatically identify themes and sentiments described by the participants. Authors KK and LW then categorised individual free-text responses under each of the main themes. Other common themes not extracted by NVivo v.12 but recognised by KK and LW are also presented. Data are presented as frequencies and percentage values. No survey weighting adjustment was conducted due to the unavailability of appropriate auxiliary variables and lack of detailed population reference data (including no data available for South Australia).

## Results

### Participant characteristics

Overall, 424 responses were received (minimum estimated response rate of 21.2%). The demographic characteristics of Fellow anaesthetists currently registered with the ANZCA are presented in Table 1 (personal communication with the ANZCA). Participants in our survey were broadly representative of Fellows registered with the ANZCA, apart from an over-representation of Fellows in VIC (41.5%) and NZ (22.6%) and an under-representation of those in NSW (16.7%) and QLD (11.1%). Males comprised 63.2% of the participants, which is consistent with data obtained from the ANZCA (68.7%). Our study participants were also representative of Fellows registered with the ANZCA in terms of age. Comparative data from the ANZCA fell within the 95% confidence interval of these values except for participants aged 60 years or more, making this age group slightly over-represented in our sample. The anonymised data is presented in S4 Appendix.

Participants were evenly distributed in terms of the number of years spent as consultant anaesthetists. Among the anaesthetists, 62.8% reported that they predominantly work in public settings, whereas 10.1% predominantly work in the private sector. In addition, 27.1% of respondents practise in both public and private settings. Most participants (85.7%) work in a metropolitan area—7.7% practise rurally and 6.7% work in both urban and rural settings. Almost all participants (98%) stated that their main area of practice is anaesthesia. Of eight participants who did not choose anaesthesia as their predominant area of practice, two participants reported that they work predominantly in intensive care and three in pain medicine.

**Table 1. Demographic features of participants (n = 424).**

| Variable | | N (%) [95% CI] | Data from ANZCA (%) |
|---|---|---|---|
| Gender | Male | 268 (63.2%) [58.4–67.8] | 68.7 |
| | Female | 139 (32.8%) | 31.3 |
| Age (years) | < 30 | 0 (0%) [0–0.9]* | 0.02 |
| | 30–39 | 93 (21.9%) [18.1–26.2] | 18.2 |
| | 40–49 | 160 (37.7%) [33.1–42.5] | 38.8 |
| | 50–59 | 97 (22.9%) [18.9–27.2] | 25.4 |
| | ≥ 60 | 57 (13.4%) [10.3–17.1] | 17.5 |
| Years of practice as an anaesthetist | < 5 | 90 (22.1%) | Not available |
| | 5–10 | 96 (23.6%) | |
| | 10–20 | 109 (26.8%) | |
| | > 20 | 112 (27.5%) | |
| Location | New South Wales | 68 (16.0%) [12.7–19.9] | 26.4 |
| | Queensland | 45 (10.6%) [7.85–13.9] | 18.3 |
| | Australia Capital Territory | 1 (0.2%) [0.01–1.3] | 1.5 |
| | Victoria | 169 (39.9%) [35.2–44.7] | 21.2 |
| | South Australia | 0 (0%) [0–0.9]* | 6.65 |
| | Western Australia | 28 (6.6) [4.4–9.4] | 9.85 |
| | Northern Territory | 4 (0.9%) [0.3–2.4] | 0.6 |
| | Tasmania | 1 (0.2%) [0.01–1.3] | 2.15 |
| | New Zealand | 92 (21.7%) [17.9–25.9] | 13.3 |

Data presented as number (proportion).

* 97.5% Confidence Interval.

The other three participants gave free-text responses stating that they work in a mixture of anaesthesia, intensive care and pain medicine or in education.

## Funeral attendance among anaesthetists

Only 25 participants (5.9% [95% CI 3.6–8.1%]) had attended a patient's funeral. However, 15.9% [95% CI 7.1–24.7%] agreed that they would be more likely to attend the funeral if the patient had died unexpectedly, while 45.4% of participants [95% CI 36.6–54.2%] remained neutral for this statement.

## Factors associated with funeral attendance

Participants' age group was the only demographic factor associated with a higher likelihood of attendance at a patient's funeral (p = 0.05). Of participants aged 50–59, 12.4% had attended a patient's funeral. For the age groups 30–39 years, 40–49 years and 60 years or older, the rate of funeral attendance was 3.2%, 4.4% and 5.3%, respectively. No significant associations between funeral attendance and years of experience, gender or geography (urban or rural) were identified.

## Formation of a special bond

Most participants (85.9%) reported that they seldom or never formed special bonds with patients or their families. Male anaesthetists were more likely than female anaesthetists to never form a special bond (15.3% vs 7.2%, p = 0.048). According to 55% of participants,

formation of a special bond with the patient or their family would make it more likely for them to attend the funeral, while 24.5% of participants remained neutral for this statement.

## Benefits of funeral attendance

Detailed responses to the benefits of anaesthetists' attendance at a patient's funeral are summarised in Table 2. Of the participants, 69.8% agreed that paying a gesture of respect to the deceased or their family would be a benefit of attending their patient's funeral. The next most commonly perceived potential benefit was expression of personal grief (32.3%). Only 5% of participants chose establishing their own professional development—making this the least commonly perceived potential benefit to themselves. Of the participants, 99 (23.3%) reported that there would be no benefit to the anaesthetist in attending the funeral. Of 85 free-text responses, the most commonly raised themes were closure of the relationship (49.4%) and relief from guilt (16.5%). Other responses included empathy for the patient and the family and avoidance of medico-legal issues.

Benefits to the family are presented in Table 2. Participants identified showing a gesture of respect to the family (65.3%) and care for the patient at the end-of-life and beyond (54.2%) as the main potential benefits to the family of their funeral attendance. Extension of the professional relationship to relatives (6.1%) was the least commonly perceived potential benefit to the family. A total of 91 participants (21.5%) reported that there would be no benefit to the family. Of 34 free-text responses, showing the family that they care for them (52.9%) and acknowledgement of the patient's death (23.5%) were the most commonly quoted potential benefits to the family. Other responses included an opportunity for the family to debrief and ask unanswered questions.

## Barriers to funeral attendance

Detailed responses about barriers to an anaesthetist's attendance at a patient's funeral are summarised in Table 3. The most commonly perceived barriers were perception that their attendance could be misinterpreted or seen as not warranted by the family (68.9%), time restraints

**Table 2. Perceived benefits to the anaesthetist and the family of attending a patient's funeral.**

| Benefits to the anaesthetist | Respondent n = 424 |
|---|---|
| Pay a gesture of respect to the deceased or their families | 296 (69.8%) |
| Express personal grief at the loss of someone you cared for | 137 (32.3%) |
| Gain a greater understanding of who the patient was before the illness | 108 (25.5%) |
| No benefit | 99 (23.3%) |
| Provide comfort and ongoing care for the bereaved family | 92 (21.7%) |
| Appear professional to the family | 63 (14.9%) |
| Establish your own professional development | 21 (5.0%) |
| **Benefits to the family** | **Respondent n = 424** |
| Pay a gesture of respect to the family | 277 (65.3%) |
| Show caring for patients at the end-of-life and beyond | 230 (54.2%) |
| Provide an opportunity for family members to ask unanswered questions | 106 (25.0%) |
| No benefit | 91 (21.5%) |
| Reduce the family's stress | 33 (7.8%) |
| Extend the relationship to relatives | 26 (6.1%) |

Participants were asked to leave this question blank if they believed that there was no benefit; multiple options could be chosen. Data presented as number (proportion).

**Table 3. Perceived barriers to anaesthetists attending a patient's funeral.**

| Barriers for anaesthetists | Respondents n = 424 |
|---|---|
| May be misinterpreted or seen as not warranted | 292 (68.9%) |
| Time restraint | 272 (64.2%) |
| It can disturb the very personal and private grieving process of a family | 255 (60.1%) |
| Presence of the anaesthetist can be traumatic for the family | 195 (46.0%) |
| Attending can invite recriminations or even anger | 154 (36.3%) |
| Attending can invite inappropriate questions | 149 (35.1%) |
| Perceived patient and/or family dissatisfaction with care | 108 (25.5%) |
| May have implications for anaesthetist–patient confidentiality | 101 (23.8%) |
| Funeral attendance is a source of emotional stress for me | 91 (21.5%) |
| Funeral attendance is unprofessional | 63 (14.9%) |
| No barrier | 40 (9.4%) |
| Personal bereavement from the loss of the patient | 37 (8.7%) |

Participants were asked to leave this question blank if they believed that there was no barrier; multiple options could be chosen. Data presented as number (proportion).

(64.2%), disruption of a private event (60.1%) and fear that their presence could be traumatic for the family (46.0%). Personal bereavement from the loss of the patient (8.7%) was the least commonly perceived barrier to attendance. Forty-five participants (9.4%) reported that there would be no barrier to attending a patient's funeral. Of 57 free-text responses, being unaware of the patient's death or not being invited to the funeral (22.8%), workload or location (17.5%) and blurring of the professional and personal barrier (17.5%) were the most frequently quoted barriers to attending a patient's funeral. Other responses included the perception that attendance is unnecessary or strange, cultural or religious differences and disapproval by colleagues.

## Subgroup analysis

Subgroup analysis of the benefits and barriers perceived by participants who had attended a patient's funeral showed similar results. Of the subgroup, 96% identified paying a gesture of respect to the deceased or their family as a perceived benefit to themselves. Only 8% of anaesthetists who had attended a patient's funeral regarded establishing their own professional development as a perceived benefit. Regarding the benefits for the family, showing care for patients at the end-of-life and beyond (96%) and paying a gesture of respect to the family (88%) were the most commonly perceived benefits. Time restraints (88%) and fear that the family may misinterpret or not warrant the attendance (76%) were major obstacles to attendance. None of the subgroup participants perceived funeral attendance as being unprofessional.

## Factors associated with perceived benefits and barriers

Older participants identified fewer potential benefits of funeral attendance for themselves; they also perceived fewer barriers to their attendance (see Table 4). Female anaesthetists were more likely to identify gaining an understanding of who the patient was before the illness as a potential benefit to attending the funeral (Odds Ratio (OR): 1.64, 95% CI: 1.04–2.59, p = 0.032). Further, they were more likely to agree that their attendance at a patient's funeral could be beneficial to show caring for patients at the end-of-life and beyond (OR: 1.79, 95% CI: 1.17–2.73, p = 0.007). Fig 1 illustrates the association between selected demographic factors and

**Table 4. Age and the expected number of perceived benefits of and barriers to attending a patient's funeral.**

|  | Age group (years) | Incidence rate ratio [95% CI] | p value |
|---|---|---|---|
| **Number of benefits** | 30–39 | 1.0 | |
|  | 40–49 | 0.83 [0.69–0.99] | 0.04 |
|  | 50–59 | 0.76 [0.61–0.94] | 0.01 |
|  | ≥ 60 | 0.71 [0.55–0.92] | 0.01 |
| **Number of barriers** | 30–39 | 1.0 | |
|  | 40–49 | 0.90 [0.80–1.01] | 0.07 |
|  | 50–59 | 0.73 [0.64–0.84] | < 0.001 |
|  | ≥ 60 | 0.69 [0.59–0.82] | < 0.001 |

The incidence rate ratio represents a factor change in the expected number of benefits and barriers chosen by the participants. The age group 30–39 is used as the reference category.

participants' perception of the total number of potential benefits of and barriers to attending their patient's funeral.

## Discussion

We performed a prospective, mixed-methods survey of practising Fellow anaesthetists in Australia and New Zealand to examine their attitudes towards attending the funeral of a patient. We found that most anaesthetists had never attended a patient's funeral. Respect, expression of grief and caring beyond life were commonly perceived potential benefits of attendance. The survey showed that few anaesthetists form close relationships with patients or their families. Families misinterpreting or not warranting their attendance and time restraints were commonly perceived barriers to funeral attendance.

### Relevance to the literature

The low attendance rate of anaesthetists at patients' funerals was highly discordant with other specialities. In Australia, specialists in other disciplines, such as palliative care, were up to 10 times more likely to attend the funerals of their patients. Zambrano et al. [20] reported that 71% of general practitioners, 67% of oncologists and psychiatrists, 63% of palliative care physicians and 52% of surgeons had attended the funeral of a patient [20]. A briefer doctor–patient relationship and fewer interactions with patients and their families in anaesthesia may explain some of our findings.

While our study showed that most anaesthetists do not form a special bond with patients or their families, the majority still perceived a close bond as a potential facilitator to attend a patient's funeral. This reiterates the findings from other studies, which suggest that feeling close to the patient or the family drives clinicians' attendance at the patient's funeral [22, 25]. This may also partially explain why the unexpected death of a patient did not influence or facilitate the anaesthetist's attendance at the funeral.

Our participants strongly believed that their attendance at a patient's funeral could be beneficial in terms of showing respect to the patient and the family. This was consistent with the findings of Senthil et al. [25] and Zambrano et al. [20], which identified showing respect for the family as one of the strongest drivers for medical practitioners' attendance at a patient's funeral. Other common themes were the expression of personal grief and showing care to the family. In contrast to some studies [9, 10], our findings did not identify professional development and an extension of the professional relationship to relatives as potential benefits. Interestingly, the number of anaesthetists who recognised no potential benefit to attending a

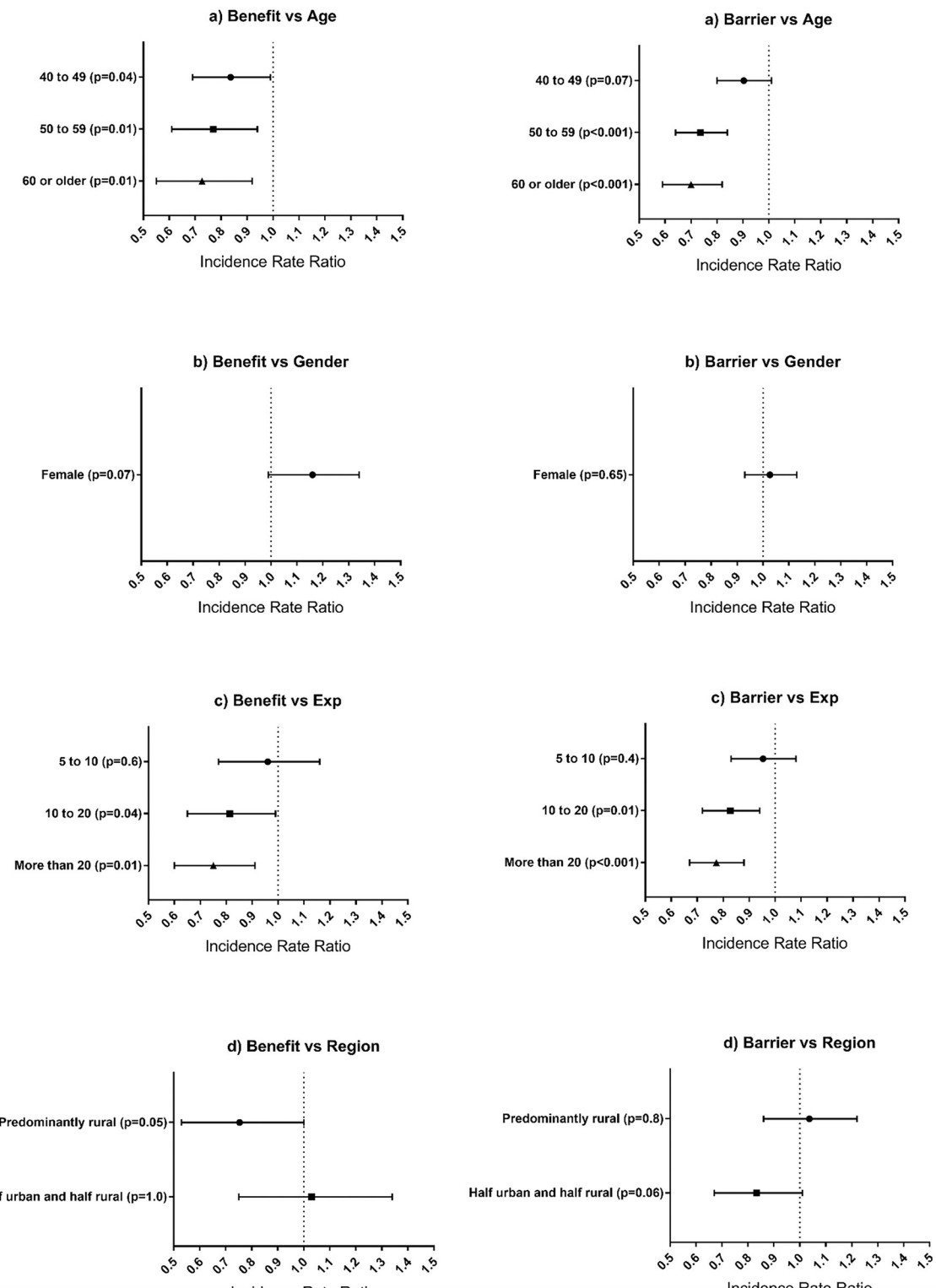

**Fig 1. Relationship between demographic factors and number of benefits of and barriers to attending their patient's funeral.** The vertical, dotted line represents the reference group and refers to the following: a) Age: 30–39, b) Male, c) Experience: Less than 5 years, d) Practice: Predominantly urban. The incidence rate ratio represents a factor change in the expected number of benefits and barriers chosen by participants compared to the reference category.

patient's funeral was double the number who perceived no barrier to attending one—perhaps alluding to anaesthetists' sceptical views towards attending a patient's funeral.

Our study showed that the most common barrier to attending a patient's funeral among our participants was the perception that the family may misinterpret the motive for attendance or not warrant their attendance. This has not been previously addressed in the current literature. Time restraint was also a significant barrier to anaesthetists' attendance at a funeral. Anaesthesia requires the continual presence of the anaesthetist in the operating room for the duration of the surgery, which may limit their flexibility to attend funerals during working hours. While this is also true for surgeons, they have more opportunities to build rapport with patients and their families during daily ward rounds and outpatient follow-ups. Results from Zambrano et al. [20] showed that more surgeons attend the funeral of their patients compared to intensivists and anaesthetists, and time restraint was not significantly associated with non-attendance. In contrast, time restraint was still a major obstacle in our study—even among anaesthetists who had attended a patient's funeral. It is unclear why there is such a disparity. Future studies could focus on comparing differences in perceived barriers to attending a patient's funeral between surgeons and anaesthetists.

Notably, the most commonly suggested barrier among the free-text responses was not being invited to the funeral and not even being aware of the patient's death. This may be explained by the brevity of the anaesthetist–patient–family relationship, unlike other specialists who form closer relationships over a longer period. It also highlights a potential disconnect of the relationship between anaesthetists and their patients after they are discharged from the post-anaesthesia care unit to the ward. This is compounded by the continuing care frequently provided postoperatively by other craft groups, such as surgeons or intensivists. Unlike anaesthetists, other medical specialists—such as general practitioners, physicians and surgeons—have greater scope in their practice for continuity of care. They are able to see patients with chronic illnesses over many years, and this reinforces the doctor–patient relationship. We speculate that this discrepancy in relationships will change in future studies as anaesthetists increasingly enter perioperative medicine.

Few participants in our study considered emotional challenges as the main barrier to their attendance at a patient's funeral. This contrasts with the findings of Borasino et al. [27], which identified personal sense of discomfort about the patient's death as the second most frequently quoted barrier among paediatric critical care specialists. Again, this illustrates the potential lack of personal bonding between anaesthetists and their patients. We also found that older and more experienced anaesthetists reported fewer barriers to attending a patient's funeral. This indicates that there may be fewer stigmas about attending a patient's funeral over time. However, our findings also revealed that these anaesthetists perceived fewer personal benefits.

It has been suggested that practitioners in small, isolated areas (such as rural communities) tend to form a closer relationship with their patients, and hence, would be more likely to attend their funeral [5, 29]. However, our results suggest that participants working in rural regions did not have a higher rate of attendance at a patient's funeral compared to those in metropolitan areas. This may be explained by our finding that 85.9% of anaesthetists rarely or never formed a special bond with patients or their families. Female anaesthetists were more open to forming a close relationship with patients or their families. They were also more likely to view funeral attendance as an opportunity to better understand and to show care for the patient. Our findings reiterate those of Zambrano et al. [20], who concluded that female practitioners are more likely to attend their patient's funeral to gain a better understanding of the patient and as a continuity of patient care. Nevertheless, our results did not show a statistically significant gender difference in funeral attendance. These findings are supported by Borasino

et al. [27], who failed to find a statistically significant gender difference for paediatric critical care specialists' rate of attendance at a patient's funeral.

## Strengths and limitations

Our study has several strengths and limitations. First, it is the largest survey to date exploring anaesthetists' attitudes towards and their perceived benefits of and barriers to attending the funeral of a patient. We had a large sample size, which was mainly representative of the population of Australian and New Zealand anaesthetists. The use of free-text responses may have allowed a more detailed qualitative analysis of some of the perceived benefits and barriers. The analysis of responses to free-text questions provided in-depth information, especially about benefits and barriers that were not presented in the survey form or the literature. The anonymous, de-identified and confidential design of the survey may have encouraged respondents to be more willing to share personal information, which might have been more challenging in a face-to-face interview setting.

As our survey was distributed by directors of anaesthesia departments in Australian and New Zealand hospitals listed on the ANZCA website, we do not know whether the survey was forwarded by all directors; therefore, the exact response rate could not be accurately determined. It is possible that our response rate is either overestimated or underestimated. As participation was completely voluntary, anaesthetists who were interested in the topic may have been more likely to be involved, which would have resulted in selection bias. However, the age and gender distribution of our participants was similar to the data provided by the ANZCA. Therefore, we believe that the risk was minimised. In addition, it is possible that participants chose more socially desirable answers that resulted in response bias. Finally, as our survey was only sent to Fellows of the ANZCA, this restricts the extrapolation of our results to anaesthesia registrars or trainees.

## Survey implications

Our findings imply that anaesthetists do not readily build rapport and develop special bonds with patients or their families. The typically short-lived professional doctor–patient relationship as well as the unique clinical setting of anaesthesia amplify this barrier. In contrast to findings from Borasino et al. [27], bereavement from the loss of the patient was the least commonly perceived barrier among our participants, suggesting the relative paucity of special bonds between patients, families and anaesthetists. In addition, the least commonly perceived potential benefit was extension of the professional relationship to relatives, again suggesting a discontinuation of the professional relationship between the anaesthetist, the patient and the patient's family post-surgery.

The ANZCA states that anaesthetists have an important and primary role in caring for the patient before, during and after surgery [30]. While this statement is open to varied interpretations, in an era where there is an emerging need for medical practitioners to employ a holistic approach to patient care, it may provide a signpost to an expanding role in the inpatient journey. The development of perioperative medicine into a speciality for anaesthetists may change this paradigm. It would allow anaesthetists to embrace the opportunities presented by the broader role of the perioperative physician, which encompasses many aspects of the 'non-operative' care of patients undergoing major surgery [31].

## Conclusion

Most anaesthetists practising in Australia and New Zealand have never attended their patient's funeral. Few anaesthetists form close relationships with patients or their families. Respect,

expression of grief and caring beyond life were commonly perceived potential benefits of attendance. Fear that families might misinterpret or not warrant their attendance and time restraints were commonly perceived barriers. Future studies could focus on the family's perspective of the anaesthetist attending the patient's funeral.

## Supporting information

**S1 Appendix. Survey template.**
(PDF)

**S2 Appendix. Invitation E-mail to directors.**
(PDF)

**S3 Appendix. Participant information and consent form.**
(PDF)

**S4 Appendix. Anonymized data file.**
(XLSX)

## Author Contributions

**Conceptualization:** Laurence Weinberg.

**Data curation:** Laurence Weinberg.

**Formal analysis:** Kwangtaek Kim, Leonid Churilov, Laurence Weinberg.

**Investigation:** Kwangtaek Kim, Rishi Mehra, Paul Anthony Stewart, Clive Rachbuch, Andrew Huang, Laurence Weinberg.

**Methodology:** Leonid Churilov, Chong Oon Tan, Roni Krieser, Rishi Mehra, Paul Anthony Stewart, Laurence Weinberg.

**Project administration:** Tuong Phan, Jake Geertsema, Roni Krieser.

**Resources:** Tuong Phan, Jake Geertsema, Roni Krieser, Rishi Mehra, Paul Anthony Stewart, Clive Rachbuch.

**Validation:** Leonid Churilov, Laurence Weinberg.

**Visualization:** Leonid Churilov, Chong Oon Tan, Laurence Weinberg.

**Writing – original draft:** Kwangtaek Kim, Chong Oon Tan, Tuong Phan, Jake Geertsema, Roni Krieser, Rishi Mehra, Paul Anthony Stewart, Clive Rachbuch, Andrew Huang, Laurence Weinberg.

**Writing – review & editing:** Kwangtaek Kim, Leonid Churilov, Chong Oon Tan, Tuong Phan, Jake Geertsema, Roni Krieser, Rishi Mehra, Paul Anthony Stewart, Clive Rachbuch, Andrew Huang, Laurence Weinberg.

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
