## [Decision Letter · Decision Letter 0]

17 Jul 2020

PONE-D-20-17956

Attitudes of anaesthetists attending the funeral of patients they care for: a cross-sectional study amongst Australian and New Zealand anaesthetists

PLOS ONE

Dear Dr. Weinberg,

Thank you for submitting your manuscript to PLOS ONE. After careful consideration, we feel that it has merit but does not fully meet PLOS ONE’s publication criteria as it currently stands. Therefore, we invite you to submit a revised version of the manuscript that addresses the points raised during the review process.

The manuscript "Attitudes of anaesthetists attending the funeral of patients they care for: a cross- sectional study amongst Australian and New Zealand anaesthetists" by Kim et al. is a well-written study that seeks to determine the attitudes of anesthetists towards attending their patients' funerals. The study was carried out via survey, has a healthy number of participants (n=424), and presents findings that are new to the literature. There are some items that need addressing, which the two reviewers have aptly noted. 

Reviewer 2 has requested some further revisions and clarification for the manuscript, which I agree with. Please address these remarks. Thank you.

We look forward to receiving your revised manuscript.

Kind regards,

Daniel Jeremiah Hurst, PhD

Academic Editor

PLOS ONE

Journal Requirements:

2. Please provide additional details regarding participant consent.

In the ethics statement in the Methods and online submission information, please ensure that you have specified (i) whether consent was informed and (ii) what type you obtained (for instance, written or verbal).

If your study included minors, state whether you obtained consent from parents or guardians.

If the need for consent was waived by the ethics committee, please include this information.

5. Please include captions for your Supporting Information files at the end of your manuscript, and update any in-text citations to match accordingly. Please see our Supporting Information guidelines for more information: http://journals.plos.org/plosone/s/supporting-information

Reviewers' comments:

Reviewer's Responses to Questions

**Comments to the Author**

1. Is the manuscript technically sound, and do the data support the conclusions?

Reviewer #1: Yes

Reviewer #2: Yes

2. Has the statistical analysis been performed appropriately and rigorously? 

Reviewer #1: Yes

Reviewer #2: Yes

3. Have the authors made all data underlying the findings in their manuscript fully available?

Reviewer #1: Yes

Reviewer #2: No

4. Is the manuscript presented in an intelligible fashion and written in standard English?

Reviewer #1: Yes

Reviewer #2: Yes

5. Review Comments to the Author

Reviewer #1: This is a useful addition to the literature. It builds on a growing literature about attendance of medical practitioners at patient's funerals. The paper is well constructed and generally well written. There are a few minor typographical errors and one or two inconsistencies within the referencing style.

Reviewer #2: Thank you for the opportunity to review the manuscript entitled: Attitudes of anaesthetists attending the funeral of patients they care for: a cross-sectional study amongst Australian and New Zealand anaesthetists.

This study addresses an important question and with a good sample size and an appropriate research design, identifies barriers and facilitators of attendance at patient funerals as well as the actual practices of anaesthetists from Australia and New Zealand.

In general, the article is well-written and most sections are well-balanced. There is however a decision regarding their analysis, which I have flagged as a concern below. In addition, I am also making some suggestions that can help improve different sections and aspects of the manuscript. Please find them below under each of the sections of the manuscript.

Introduction

- Could you please elaborate in a sentence or two, why is it important to study the perspective of anaesthetists? This comes later on in the discussion, but already a justification in this section would be important.

Methods

- I suggest that you employ either the word ‘survey’ or the word ‘questionnaire’, but not that you use it interchangeably. An example of why this can be confusing is on the ‘study design’ section in lines 88 and 89 where it is unclear whether they are the same, or whether there was a survey and a separate questionnaire.

- The ‘study design’ paragraph, particularly the description of the survey is confusing. Starting on line 88, the 17 questions are introduced, with an overview of the first 10 questions which were about barriers, attitudes and benefits. Then a general statement of the formation of bonds, and information about those being measured on a likert scale, then it restates on line 93 that barriers and attitudes were assessed. This paragraph could be better organised to describe the areas addressed and the types of questions asked in a clearer way.

- What was the result of the pilot testing? Did it lead to reformulating questions, etc? Otherwise, why were those responses not used, as stated on page 6, line 124? It would be interesting to elaborate on this in a sentence or two.

- A better description of ‘how’ and ‘who’ extracted the ‘main themes from free text responses’ is needed, first to understand the rigour of this data extraction, but also to understand whether actual themes were extracted. Can it really be called ‘thematic analysis’?

Results

- I find it interesting that for benefits and barriers only the smaller proportion of participants who provided a response is reported, when it is stated that a blank response was also interpretable, i.e. as identifying no benefits or no barriers. E.g. In benefits to anesthetists, the results are based on 325 respondents, meaning that at least 25% of their sample identified no benefit whatsoever. Shouldn’t this have been incorporated into the results, so that the 91% who identify at least one benefit (in this example) are not over inflated?

- Similarly, for the factors associated with perceived benefits and barriers (starting on line 213), can you elaborate on how you handled the nonresponses as they are not skipped questions. They are not used in the descriptive part, but it is not clear whether they have been used in the analytical tests, can you please explain how did you use them / not use them and what was the rationale?

- The paragraph describing participant characteristics and Table 1 present the exact same information for location and age distribution. Please shorten the paragraph by pinpointing only the most relevant aspects of those characteristics and avoid presenting the CIs on the text, as they are all in the table and the parenthesis affect readability.

- It is established that only 6% of respondents have attended a patient funeral, yet when discussing barriers, bonds, and benefits, most of the sentences describe those attitudes as if they were actually occurring. E.g. “the most commonly perceived barriers were perception that their attendance is misinterpreted”, this should read instead: “could be misinterpreted”. Another example: “participants identified (…) caring for the patient beyond life (69%) as the main benefits of their attendance at a patient’s funeral”, this sentence should highlight that these are the main benefits of their “potential attendance”. Another example: “formation of a special bond (…) makes it more likely for them to attend the funeral” should be “would make it more likely” as you are reporting on the whole sample. Unless this data is from the 6% who do attend funerals (which would need to be stated) please highlight them as hypothetical using words such as ‘potential’ or “would attend” or ‘could be reasons’ as a way to highlight this difference between potential reasons for a behaviour and not the reasons for their actual behaviour, as it currently reads. Even better, you could also make a differentiation and highlight in the different sections what were the benefits, barriers, etc. for the 6% who attend funerals and in this way help the reader have more clarity when reading and interpreting your findings.

- The description of Table 3 says on line 210 “participants were asked to leave this question blank if they believed that there is no benefit”, since this table is about barriers, please replace the word 'benefit' for ‘barrier’.

- With regards to presentation of results, all throughout the results section, please avoid using numbers when you start a sentence. Either start with words, or spell out the number (e.g. do not start the sentence with: 21.0% and instead start it with: Twenty-one percent).

Discussion

- On line 266, you state that a reason why anaesthetists perceive time restraint as a reason not to attend funerals is because they have to be in the OR for the entire duration of the surgery. Wouldn’t this hold true for surgeons too, who you highlight earlier that have been found to attend funerals in other studies?

- Sentence starting on line 321 is unclear. What is meant by emotional challenges, can you be more explicit?

6. PLOS authors have the option to publish the peer review history of their article (what does this mean?). If published, this will include your full peer review and any attached files.

Reviewer #1: No

Reviewer #2: **Yes: **Sofia Zambrano

---

## [Author Response · Author response to Decision Letter 0]

1 Sep 2020

We would like to thank the Editorial team for considering our manuscript for publication and providing us a valuable opportunity to revise the manuscript.

1. Editorial request: Please ensure that your manuscript meets PLOS ONE's style requirements, including those for file naming.

Authors’ response: Please accept our apologies for the oversight. The manuscript has now been reformatted in accordance to PLOS ONE’s style requirements, including the name of the files. In addition, we have ensured all references conform to the PLOS One style. 

2. Editorial request: Please provide additional details regarding participant consent. In the ethics statement in the Methods and online submission information, please ensure that you have specified (i) whether consent was informed and (ii) what type you obtained (for instance, written or verbal). If your study included minors, state whether you obtained consent from parents or guardians. If the need for consent was waived by the ethics committee, please include this information.

Authors’ response: Thank you for this important comment regarding participant consent. Our invitation E-mail that was distributed to participants included a summary and objectives of the study and a written information about consent. The participant consent information sheet stated that participation was completely voluntary, and consent was implied upon completion of the survey. In addition, the consent information sheet stated that if the participant decided not to take part in the survey, it will not impact upon their employment with their current health organisation or their standing with ANZCA. We have added the participant information and consent form as Appendix 3. Further, we have added more details regarding participant consent into the revised manuscript. The updated relevant section now states:

“An invitation email was sent directly to the directors of anaesthesia departments in Australian and New Zealand hospitals listed on the ANZCA website (http://www.anzca.edu.au). This invitation email included the objectives and a summary of the study, written information about consent (which stated that consent was assumed on completion of the survey as participation was completely voluntary) and a link to the online survey. A copy of the invitation email is presented in S2 Appendix. The participant information and consent form are presented in S3 Appendix.”

3. Editorial request: We note that you have indicated that data from this study are available upon request. PLOS only allows data to be available upon request if there are legal or ethical restrictions on sharing data publicly. 

b) If there are no restrictions, please upload the minimal anonymized data set necessary to replicate your study findings as either Supporting Information files or to a stable, public repository and provide us with the relevant URLs, DOIs, or accession numbers. Please see https://clicktime.symantec.com/3F1eMVLX1LecNR1SHkazyT17Vc?u=http%3A%2F%2Fwww.bmj.com%2Fcontent%2F340%2Fbmj.c181.long for guidelines on how to de-identify and prepare clinical data for publication. For a list of acceptable repositories, please see https://clicktime.symantec.com/3a3YipzaaAqF6KuFsuKCYH7Vc?u=http%3A%2F%2Fjournals.plos.org%2Fplosone%2Fs%2Fdata-availability%23loc-recommended-repositories.

Authors’ response: Thank you for this important comment regarding our data. There are no restrictions on our data access and we have now uploaded our anonymized data set as a Supporting Information file.

4. Editorial request: Please amend your list of authors on the manuscript to ensure that each author is linked to an affiliation. Authors’ affiliations should reflect the institution where the work was done (if authors moved subsequently, you can also list the new affiliation stating “current affiliation:….” as necessary)

Authors’ response: Thank you for this comment. We have updated each author’s affiliation to reflect where the work was done. 

5. Editorial request: Please include captions for your Supporting Information

files at the end of your manuscript, and update any in-text citations to match accordingly. Please see our Supporting Information guidelines for more information: http://journals.plos.org/plosone/s/supporting-information

Authors’ response: Thank you for the comment. We have included captions for Supporting Information files 1, 2 and 3 at the end of the manuscript. 

Academic Reviewers queries

We would like to thank the expert academic reviewers for their constructive and insightful comments and for their time in reviewing our manuscript. We are very grateful for all the positive comments we have received from reviewer’s Dr Sofia Zambrano and Reviewer #1. Over the last few years, we have been inspired by the research that Dr. Zambrano has conducted, particularly the research pertaining to doctor-patient relationships and bereavement practices employed by Australian medical practitioners, including attendance at their patient’s funeral. In part, this has been one of the motivating drivers that inspired our group to undertake this research. Dr Zambrano’s insightful comments have enabled us to strengthen the scientific rigor and merits of our research.

Reviewer #1: Queries

We thank expert Reviewer#1 for reviewing our manuscript. We genuinely appreciate the expert comments provided.

1. Expert reviewer: This is a useful addition to the literature. It builds on a growing literature about attendance of medical practitioners at patient's funerals. The paper is well constructed and generally well written. There are a few minor typographical errors and one or two inconsistencies within the referencing style.

Authors’ response: Thank you for these positive comments. We have carefully reviewed the manuscript and corrected all typographical errors. In addition, we have revised all the references to ensure they align with the required style and formatting. We believe that the results of our manuscript will become increasingly relevant as the role of anaesthetists in perioperative care expands. Thank you once again for taking the time to review our manuscript. 

Reviewer 2: Dr. Sofia Zambrano’s queries

We thank expert Dr. Zambrano for taking the time to review our manuscript. We have been inspired by the research that Dr. Zambrano has conducted, particularly the research pertaining to doctor-patient relationships and bereavement practices employed by Australian medical practitioners, including attendance at their patient’s funeral. In part, this has been one of the motivating drivers for our research team to undertake research in this area. We genuinely appreciate the expert comments provided. 

1. Expert reviewer: Thank you for the opportunity to review the manuscript entitled: Attitudes of anaesthetists attending the funeral of patients they care for: a cross-sectional study amongst Australian and New Zealand anaesthetists. This study addresses an important question and with a good sample size and an appropriate research design, identifies barriers and facilitators of attendance at patient funerals as well as the actual practices of anaesthetists from Australia and New Zealand. In general, the article is well-written and most sections are well-balanced. There is however a decision regarding their analysis, which I have flagged as a concern below. In addition, I am also making some suggestions that can help improve different sections and aspects of the manuscript. Please find them below under each of the sections of the manuscript.

Authors’ response: We thank Dr. Zambrano for these positive comments. We appreciate the insightful and expert suggestions provided. The constructive comments have also helped us improve our manuscript by enhancing its academic merit and scientific rigor. We have addressed all the constructive suggestions in our responses below.

2. Expert reviewer: Introduction: Could you please elaborate in a sentence or two, why is it important to study the perspective of anaesthetists? This comes later on in the discussion, but already a justification in this section would be important.

Authors’ response: Thank you for this excellent comment. We agree that it is crucial to explain why a specific group was studied early in the manuscript so that the audience can read with a context. 

In the manuscript we have now included the following statement in the revised manuscript (Background section, second paragraph):

“With an increasing focus on perioperative medicine in anaesthesia training, anaesthetists are increasingly responsive to the preferences, needs and values of the individual patient. With more anaesthetists embracing perioperative medicine as a distinct subspecialty, a deeper understanding of what is important to the patient is being developed. This understanding fosters trust, establishes mutual respect and facilitates a unique clinician–patient-centred approach to shared planning and decision-making. Hence, anaesthetists currently have more opportunities to develop a strong rapport with patients and their families. Accordingly, over time, more anaesthetists will either be interested or be asked to attend the funeral of their patients. However, the attitudes of other medical specialists towards funeral attendance remains largely unexplored, and there is no large study specifically exploring the attitudes, benefits and barriers of attending a patient’s funeral as perceived by anaesthetists.”

3. Expert reviewer: Methods: I suggest that you employ either the word ‘survey’ or the word ‘questionnaire’, but not that you use it interchangeably. An example of why this can be confusing is on the ‘study design’ section in lines 88 and 89 where it is unclear whether they are the same, or whether there was a survey and a separate questionnaire.

Authors’ response: Thank you for this excellent suggestion. We agree that using the word ‘survey’ and the word’ questionnaire’ can create confusion to the reader. The word “questionnaire” has been changed to “survey” throughout the manuscript. 

4. Expert reviewer: Methods: The ‘study design’ paragraph, particularly the description of the survey is confusing. Starting on line 88, the 17 questions are introduced, with an overview of the first 10 questions which were about barriers, attitudes and benefits. Then a general statement of the formation of bonds, and information about those being measured on a likert scale, then it restates on line 93 that barriers and attitudes were assessed. This paragraph could be better organised to describe the areas addressed and the types of questions asked in a clearer way.

Authors’ response: Thank you for your suggestion for improvement. After reviewing the ‘Study design’ paragraph, we agreed that sentences were not well interlinked, and the reader would have a difficulty knowing how each survey question was structured. We have amended the manuscript such that the format of survey questions are better described, and the aim of each question is more clearly explained. The revised paragraph reads now states:

The first 10 questions explored participants’ previous experience of attending a patient’s funeral, frequency of formation of special bonds with patients or their families, factors that affect their attendance at a patient’s funeral and perceived benefits of and barriers to attending a patient’s funeral. Having a special bond with patients or their families and the unexpected death of a patient were the two factors that were investigated in the survey. A 5-point Likert scale (strongly agree to strongly disagree) was used to determine if these two factors affected participants’ attendance at their patient’s funeral. Survey questions that explored perceived benefits of and barriers to attending a patient’s funeral allowed participants to choose none, one or multiple options. There were also three open-ended questions where participants could provide free-text responses regarding the benefits of and barriers to their attendance at a patient’s funeral. The next seven questions explored participants’ geographic region, age, gender, type of hospital (rural or urban), their area of practice as well as whether they mainly practise in a public or private setting. A copy of the survey is presented in S1 Appendix.

5. Expert reviewer: Methods: What was the result of the pilot testing? Did it lead to reformulating questions, etc.? Otherwise, why were those responses not used, as stated on page 6, line 124? It would be interesting to elaborate on this in a sentence or two.

Authors’ response: Thank you for this important question regarding the pilot study participants. The purpose of the pilot study was to ascertain that the survey questions were easy to understand and not misleading. An important factor was to ensure that the questionnaire items accurately addressed each of our research questions and tested whether the questionnaire was comprehensible and appropriate, and that the questions were well-defined, clearly understood and presented in a consistent manner. Pilot study participants were recruited via direct personal communications. Our pilot testing did not lead to the reformulating of any questions, only to the correction to minor syntax and grammar. As such, survey questions used in the formal study differed on slightly from those used in the pilot study. There were no additional questions included, and no questions were excluded. 

Given that open and reflective discussions about the study, its aims, objectives, and design occurred between the researchers and the pilot study participants, we were concerned that this may have introduced a tendency towards a more prejudiced viewpoint; therefore to avoid this bias, we excluded the pilot study participants from being included in the main study. This has been included in the revised manuscript.

We have amended the manuscript stating the reasons why the results of the pilot study were not used. We have re-written the ‘Study design’ section to now state:

“After a literature review of anaesthetists’ views on attending a patient’s funeral, we designed and developed an online survey using commercial software (SurveyMonkey Inc., San Mateo, California, USA). The survey was pilot tested on 10 anaesthetists from rural, secondary and tertiary level hospitals to verify whether the survey questions were comprehensible, appropriate, well-defined and not misleading. We asked whether the questions were presented in a consistent manner. Responses from the pilot testing did not lead to reformulation of any questions. No additional questions were included and no questions were excluded. Minor corrections to syntax and grammar were made, and the final survey questions differed only slightly from the pilot questionnaire. The final survey consisted of 17 questions. Given that open and reflective discussions about the study, its aims, objectives and design occurred between the researchers and the pilot study participants, a tendency towards a more prejudiced viewpoint could have been introduced; therefore, to avoid this bias, the pilot study participants were excluded from participating in the final survey.

6. Expert reviewer: Methods: A better description of ‘how’ and ‘who’ extracted the ‘main themes from free text responses’ is needed, first to understand the rigour of this data extraction, but also to understand whether actual themes were extracted. Can it really be called ‘thematic analysis’?

Authors’ response: Thank you for this comment. NVivo v.12 was used as a scanning tool to extract main themes derived from free text responses. The powerful and automated processing supported our structured yet broad-brush analysis by automatically coding the sets of data. In turn, we gained deeper insights from the data by being able to automatically identify themes and sentiments described by the participants. Author KK categorised each free text response into the main themes derived from NVivo. Other themes or common responses not recognised by NVivo but deemed important by the other authors are also presented in the manuscript. We agree that the description of how the thematic analysis was performed could be improved. We have updated the manuscript to reads as follows: 

“To extract the main themes from free-text responses, NVivo v.12 was employed. This powerful and automated process supported a structured analysis by automatically coding the sets of data. In turn, we gained deeper insights from the data by being able to automatically identify themes and sentiments described by the participants. Authors KK and LW then categorised individual free-text responses under each of the main themes. Other common themes not extracted by NVivo v.12 but recognised by KK and LW are also presented.”

7. Expert reviewer: Results: I find it interesting that for benefits and barriers only the smaller proportion of participants who provided a response is reported, when it is stated that a blank response was also interpretable, i.e. as identifying no benefits or no barriers. E.g. In benefits to anesthetists, the results are based on 325 respondents, meaning that at least 25% of their sample identified no benefit whatsoever. Shouldn’t this have been incorporated into the results, so that the 91% who identify at least one benefit (in this example) are not over inflated?

Authors’ response: Thank you for this insightful comment. Upon reviewing the mentioned paragraph, the authors agree that the numbers may appear over-inflated as non-respondents were omitted from our calculations. As you have corrected indicated, a significant portion of participants found no potential benefits or barriers to attending a patient’s funeral. After discussions among the authors, we decided to add non-respondents (i.e. no benefits or barriers) as a separate category. Subsequently, for clarity and we have unified the denominator to 424 (total number of respondents) and changed percentage values accordingly. In addition, we have added non-respondents (no benefits or barriers) as a separate category in Tables 2 and 3. We hope this improves the clarity of our findings.

We have included the updated Tables in our resubmission manuscript. 

8. Expert reviewer: Results: Similarly, for the factors associated with perceived benefits and barriers (starting on line 213), can you elaborate on how you handled the nonresponses as they are not skipped questions. They are not used in the descriptive part, but it is not clear whether they have been used in the analytical tests, can you please explain how did you use them / not use them and what was the rationale?

Authors’ response: Thank you for this important question. The paragraph pertaining to the subheading ‘Factors associated with perceived benefits and barriers’ contain statistical analysis investigating the relationship with age group and the number of perceived benefits of and barriers to attending a patient’s funeral. Therefore, non-respondents were included in the analysis. For example, if a participant left the barrier question blank, this would have been interpreted as zero number of barriers to attending a patient’s funeral. As illustrated in Table 4, participants in older age groups selected fewer number of benefits and barriers. Whilst we did not specifically look at the data, this could mean that more participants in older age group left the benefits and barriers questions blank. 

9. Expert reviewer: Results: The paragraph describing participant characteristics and Table 1 present the exact same information for location and age distribution. Please shorten the paragraph by pinpointing only the most relevant aspects of those characteristics and avoid presenting the CIs on the text, as they are all in the table and the parenthesis affect readability.

Authors’ response: Thank you for your suggestion. We agree that participant characteristics section may impair the readability especially because of confidence intervals are double presented. We have amended the manuscript such that the paragraph under participants characteristics only contains sentences pertaining to the comparison between our study participants and data from ANZCA. We have left the latter half of the paragraph relatively untouched as it presents important data that is not presented in Table 1. 

The paragraph under the participants characteristics section now read as follows:

“Overall, 424 responses were received (minimum estimated response rate of 21.2%). The demographic characteristics of Fellow anaesthetists currently registered with the ANZCA are presented in Table 1 (personal communication with the ANZCA). Participants in our survey were broadly representative of Fellows registered with the ANZCA, apart from an over-representation of Fellows in VIC (41.5%) and NZ (22.6%) and an under-representation of those in NSW (16.7%) and QLD (11.1%). Males comprised 63.2% of the participants, which is consistent with data obtained from the ANZCA (68.7%). Our study participants were also representative of Fellows registered with the ANZCA in terms of age. Comparative data from the ANZCA fell within the 95% confidence interval of these values except for participants aged 60 years or more, making this age group slightly over-represented in our sample.

Participants were evenly distributed in terms of the number of years spent as consultant anaesthetists. Among the anaesthetists, 62.8% reported that they predominantly work in public settings, whereas 10.1% predominantly work in the private sector. In addition, 27.1% of respondents practise in both public and private settings. Most participants (85.7%) work in a metropolitan area—7.7% practise rurally and 6.7% work in both urban and rural settings. Almost all participants (98%) stated that their main area of practice is anaesthesia. Of eight participants who did not choose anaesthesia as their predominant area of practice, two participants reported that they work predominantly in intensive care and three in pain medicine. The other three participants gave free-text responses stating that they work in a mixture of anaesthesia, intensive care and pain medicine or in education.”

10. Expert reviewer: Results: It is established that only 6% of respondents have attended a patient funeral, yet when discussing barriers, bonds, and benefits, most of the sentences describe those attitudes as if they were actually occurring. E.g. “the most commonly perceived barriers were perception that their attendance is misinterpreted”, this should read instead: “could be misinterpreted”. Another example: “participants identified (…) caring for the patient beyond life (69%) as the main benefits of their attendance at a patient’s funeral”, this sentence should highlight that these are the main benefits of their “potential attendance”. Another example: “formation of a special bond (…) makes it more likely for them to attend the funeral” should be “would make it more likely” as you are reporting on the whole sample. Unless this data is from the 6% who do attend funerals (which would need to be stated) please highlight them as hypothetical using words such as ‘potential’ or “would attend” or ‘could be reasons’ as a way to highlight this difference between potential reasons for a behaviour and not the reasons for their actual behaviour, as it currently reads. Even better, you could also make a differentiation and highlight in the different sections what were the benefits, barriers, etc. for the 6% who attend funerals and in this way help the reader have more clarity when reading and interpreting your findings.

Authors’ response: Thank you for this attentive comment. As you correctly indicated, only a small proportion of participants have attended their patient’s funeral. We agree that the sentences should have been more explicit in stating that these are potential benefits and barriers. 

We have now modified the paragraph as you kindly suggested by using words such as “would be”, “could be” and “potential”. In addition, we have performed a subgroup analysis on participants who have attended their patient’s funeral thanks to your advice. Interestingly, subgroup analysis showed similar results. We have added the information in a new paragraph under the subheading ‘Subgroup analysis’ which reads as follows: 

“Subgroup analysis of the benefits and barriers perceived by participants who had attended a patient’s funeral showed similar results. Of the subgroup, 96% identified paying a gesture of respect to the deceased or their family as a perceived benefit to themselves. Only 8% of anaesthetists who had attended a patient’s funeral regarded establishing their own professional development as a perceived benefit. Regarding the benefits for the family, showing care for patients at the end-of-life and beyond (96%) and paying a gesture of respect to the family (88%) were the most commonly perceived benefits. Time restraints (88%) and fear that the family may misinterpret or not warrant the attendance (76%) were major obstacles to attendance. None of the subgroup participants perceived funeral attendance as being unprofessional.”

11. Expert reviewer: Results: The description of Table 3 says on line 210 “participants were asked to leave this question blank if they believed that there is no benefit”, since this table is about barriers, please replace the word 'benefit' for ‘barrier’.

Authors’ response: Please accept our apologies for the error and thank you for pointing it out. We have now changed the word ‘benefit’ to ‘barrier’. 

12. Expert reviewer: Results: With regards to presentation of results, all throughout the results section, please avoid using numbers when you start a sentence. Either start with words or spell out the number (e.g. do not start the sentence with: 21.0% and instead start it with: Twenty-one percent).

Authors’ response: Thank you for the thoughtful comment. We agree that starting a sentence with a number may impair readability and ensured that the numbers are spelt out if they come at the start of the sentence. 

13. Expert reviewer: Discussion: On line 266, you state that a reason why anaesthetists perceive time restraint as a reason not to attend funerals is because they have to be in the OR for the entire duration of the surgery. Wouldn’t this hold true for surgeons too, who you highlight earlier that have been found to attend funerals in other studies?

Authors’ response: Thank you for your question. As you correctly pointed out, both anaesthetists and surgeons need to be present in the theatre for the entire duration of the operation. As identified in your publication ‘Attending patient funerals: Practices and attitudes of Australian medical practitioners’ published in Death Studies in 2017, 52% of surgeons have attended their patient’s funeral and time restraint was not a significant barrier. Interestingly, however, our subgroup analysis showed that even anaesthetists who have attended their patient’s funeral identified time restraint as a significant barrier. We think this disparity may originate from the relative paucity of anaesthetist-patient-family rapport compared to surgeons. As opposed to surgeons who have more opportunity to build a special bond with patients and families during daily ward rounds, anaesthetists do not routinely encounter patients postoperatively. As a result, more surgeons tend to attend their patient’s funeral. We believe that investigating the underlying reason for such contrast would be an interesting starting point for future researches. Relevant parts of the manuscript now states:

“Our study showed that the most common barrier to attending a patient’s funeral among our participants was the perception that the family may misinterpret the motive for attendance or not warrant their attendance. This has not been previously addressed in the current literature. Time restraint was also a significant barrier to anaesthetists’ attendance at a funeral. Anaesthesia requires the continual presence of the anaesthetist in the operating room for the duration of the surgery, which may limit their flexibility to attend funerals during working hours. While this is also true for surgeons, they have more opportunities to build rapport with patients and their families during daily ward rounds and outpatient follow-ups. Results from Zambrano et al. [20] showed that more surgeons attend the funeral of their patients compared to intensivists and anaesthetists, and time restraint was not significantly associated with non-attendance. In contrast, time restraint was still a major obstacle in our study—even among anaesthetists who had attended a patient’s funeral. It is unclear why there is such a disparity. Future studies could focus on comparing differences in perceived barriers to attending a patient’s funeral between surgeons and anaesthetists.”

14. Expert reviewer: Discussion: Sentence starting on line 321 is unclear. What is meant by emotional challenges, can you be more explicit?

Authors’ response: Please accept our apologies for the unclarity. After reviewing, authors agreed that the purpose of the paragraph was to highlight the poverty of rapport between anaesthetists, patients and families due to a shorter-lived professional relationship. By mentioning emotional challenges, we were pertaining to the finding where ‘personal bereavement from the loss of the patient’ was the least commonly perceived barrier in our study. Unfortunately, the original sentence was not written correctly, and the reader would have interpreted that such emotional challenge was a benefit to attending the patient’s funeral. We have amended the manuscript to better deliver our message, and it reads as follows:

“Our findings imply that anaesthetists do not readily build rapport and develop special bonds with patients or their families. The typically short-lived professional doctor–patient relationship as well as the unique clinical setting of anaesthesia amplify this barrier. In contrast to findings from Borasino et al. [27], bereavement from the loss of the patient was the least commonly perceived barrier among our participants, suggesting the relative paucity of special bonds between patients, families and anaesthetists. In addition, the least commonly perceived potential benefit was extension of the professional relationship to relatives, again suggesting a discontinuation of the professional relationship between the anaesthetist, the patient and the patient’s family post-surgery.”

Once again, thank you for taking the time to review and consider our manuscript for publication in PLOS ONE. The comprehensive reviews and constructive comments provided by yourself and the expert reviewers have been immensely appreciated. 

A/Prof Laurence Weinberg

BSc, MBBCh,MRCP,DPCritCareEcho,FANZCA,MD

Director, Department of Anesthesia, Austin Hospital

Associate Professor, Department of Surgery, University of Melbourne

Associate Professor, Perioperative Pain and Medicine Unit, Department of Surgery, University of Melbourne

---

## [Editor Report · Decision Letter 1]

17 Sep 2020

Anaesthetists’ attitudes towards attending the funerals of their patients: A cross-sectional study among Australian and New Zealand anaesthetists

PONE-D-20-17956R1

Dear Dr. Weinberg,

We’re pleased to inform you that your manuscript has been judged scientifically suitable for publication and will be formally accepted for publication once it meets all outstanding technical requirements.

Kind regards,

Daniel Jeremiah Hurst, PhD

Academic Editor

PLOS ONE

Additional Editor Comments (optional):

Thank you for your detailed explanations to the queries.
---

## [Editor Report · Acceptance letter]

26 Oct 2020

PONE-D-20-17956R1 

Anaesthetists’ attitudes towards attending the funerals of their patients: A cross-sectional study among Australian and New Zealand anaesthetists 

Dear Dr. Weinberg:

I'm pleased to inform you that your manuscript has been deemed suitable for publication in PLOS ONE. Congratulations! Your manuscript is now with our production department. 

Kind regards, 

on behalf of

Dr. Daniel Jeremiah Hurst 

Academic Editor

PLOS ONE